# Recent Insights into the Pathogenesis of Acute Porphyria Attacks and Increasing Hepatic PBGD as an Etiological Treatment

**DOI:** 10.3390/life12111858

**Published:** 2022-11-11

**Authors:** Daniel Jericó, Karol M. Córdoba, Ana Sampedro, Lei Jiang, Gilles Joucla, Charlotte Cabanne, José Luis Lanciego, Paolo G. V. Martini, Pedro Berraondo, Matías A. Ávila, Antonio Fontanellas

**Affiliations:** 1Hepatology Program, Center for Applied Medical Research (CIMA), University of Navarra, 31008 Pamplona, Spain; 2Navarra Institute for Health Research (IDISNA), 31008 Pamplona, Spain; 3Moderna Inc., Cambridge, MA 02139, USA; 4Structure and Activity of Biological Macromolecules, Institute of Chemistry and Biology of Membranes and Nano-Objects, ENSTBB, Bordeaux INP, 33076 Bordeaux, France; 5Neurosciences Department, Center for Applied Medical Research (CIMA), University of Navarra, 31008 Pamplona, Spain; 6Centro de Investigación Biomédica en Red de Enfermedades Neurodegenerativas (CIBERNed), Instituto de Salud Carlos III, 28031 Madrid, Spain; 7Program of Immunology and Immunotherapy, Center for Applied Medical Research (CIMA), University of Navarra, 31008 Pamplona, Spain; 8Centro de Investigación Biomédica en Red de Cáncer (CIBERONC), Instituto de Salud Carlos III, 28029 Madrid, Spain; 9Centro de Investigación Biomédica en Red de Enfermedades Hepáticas y Digestivas (CIBERehd), Instituto de Salud Carlos III, 28029 Madrid, Spain

**Keywords:** rare metabolic diseases, hemoproteins, liver function, mitochondrial cytochromes, antioxidant and detoxifying responses, glucose homeostasis, systemic messenger RNA therapy, AAV-mediated liver-directed gene therapy, enzyme replacement therapy

## Abstract

Rare diseases, especially monogenic diseases, which usually affect a single target protein, have attracted growing interest in drug research by encouraging pharmaceutical companies to design and develop therapeutic products to be tested in the clinical arena. Acute intermittent porphyria (AIP) is one of these rare diseases. AIP is characterized by haploinsufficiency in the third enzyme of the heme biosynthesis pathway. Identification of the liver as the target organ and a detailed molecular characterization have enabled the development and approval of several therapies to manage this disease, such as glucose infusions, heme replenishment, and, more recently, an siRNA strategy that aims to down-regulate the key limiting enzyme of heme synthesis. Given the involvement of hepatic hemoproteins in essential metabolic functions, important questions regarding energy supply, antioxidant and detoxifying responses, and glucose homeostasis remain to be elucidated. This review reports recent insights into the pathogenesis of acute attacks and provides an update on emerging treatments aimed at increasing the activity of the deficient enzyme in the liver and restoring the physiological regulation of the pathway. While further studies are needed to optimize gene therapy vectors or large-scale production of liver-targeted PBGD proteins, effective protection of PBGD mRNA against the acute attacks has already been successfully confirmed in mice and large animals, and mRNA transfer technology is being tested in several clinical trials for metabolic diseases.

## 1. Introduction

Heme is a porphyric ring composed of ferrous iron (Fe^2+^) and protoporphyrin IX. Although it is produced in all nucleated cells, the bone marrow and liver are the main organs for heme synthesis, producing 80% and 15% of total heme, respectively [1]. In the liver, heme is an essential component of hemoproteins which participates in the removal of waste products and poisonous substances from the blood, regulating glucose homeostasis, and facilitating the antioxidant response, cell proliferation, and energy supply of cells through the cytochromes of the mitochondrial respiratory chain.

Maintaining an appropriate intracellular heme level is crucial, since excess heme is toxic, and its deficiency is detrimental to cell metabolism. The first enzyme of the pathway, δ-aminolevulinic acid synthase (ALAS, EC 2.3.1.37), tightly regulates heme biosynthesis (Figure 1). However, loss-of-function mutations in any of the following seven enzymes cause specific metabolic disturbances, which contribute to a heterogeneous group of orphan diseases called porphyrias (Table 1) [2,3,4]. The management of porphyrias is challenging as their pathogenesis is not sufficiently understood and their treatment is still an unmet medical need because current drugs do not fully restore the disease in either biochemical or clinical terms. The knowledge generated in reference centers is of great value to better describe the natural history of rare diseases and to leverage the efforts of pharmaceutical companies in the design and development of innovative orphan drugs to be tested in the clinical arena.

## 2. Acute Intermittent Porphyria

Acute intermittent porphyria (AIP) is caused by haploinsufficiency of porphobilinogen deaminase (PBGD) and is characterized by disabling neurovisceral attacks and chronic disease symptoms. The prevalence of AIP is estimated to be 5–10 per million in the US, UK, and western Europe. In Sweden (100 per million), and in two valleys in the Murcia region in Spain (53.8 per million), it appears with a very high prevalence due to founder mutations [8]. However, the prevalence of genetic defects in the general population is much higher (1 in 1780 Caucasian individuals in USA [9], or 1 in 1675 in a study of the French population [10]), implying low penetrance of the disease.

In classical AIP, both the non-erythroid and erythroid-specific enzymes have reduced activity (50%), whereas in the so-called variant AIP, the enzymatic defect is present only in non-erythroid cells and is caused by defects in exon 1. Treatment of patients with the classical or erythroid AIP variant is based on the restoration of the liver heme synthesis pathway, since this organ is the main source of the toxic porphyrin precursors associated with the pathogenesis of acute attacks: δ-aminolevulinic acid (ALA) and porphobilinogen (PBG). This therapeutically relevant notion is supported by experimental and clinical evidence that is explained below.

Bone marrow transplantation restored erythrocyte PBGD activity in AIP mice, emulating the AIP variant [11]. However, phenobarbital administration in these mice reproduced key features of acute attacks, such as a massive increase in urinary porphyrin precursors excretion and impaired motor coordination. In humans, complete biochemical and symptomatic resolution of AIP was observed in all patients after orthotopic liver transplantation (OLT) [12]. In contrast, domino liver transplantation of AIP livers was sufficient to cause acute attacks in nonporphyric recipients with normal heme synthesis in the other organs [13]. Therefore, these data point to the liver as the major etiologic site of this disorder.

### Acute Neurovisceral Attacks

A secondary up-regulation of the first and rate-limiting enzyme in hepatic heme synthesis ALAS1 results in an overproduction of the potentially neurotoxic heme precursors ALA and PBG, closely associated with neurovisceral attacks. Indeed, the first acute attack emerges in most patients after exposure to precipitating factors such as drugs or other chemicals (Drugs database: http://www.drugs-porphyria.org, accessed on 11 November 2022), alcohol intake, acute illness, infection, stress, physical exhaustion, calorie deprivation, and steroid hormones, mainly oestrogens and progesterone, that regulate the reproductive cycle in women [2,3]. All these triggering factors have in common the induction of liver ALAS1 mRNA expression either directly through the peroxisome proliferator-activated receptor-gamma coactivator (PGC1α, e.g., fasting) [14] by positive feedback caused by excessive heme consumption to form hemoproteins (for example, CYP450), or by increasing its degradation through the induction of heme oxygenase-1 (HO-1), the key enzyme of heme catabolism [15,16]. HO-1 is up-regulated in the event of the hypoxia, inflammation, or oxidative stress associated with acute illness, infection, stress, or physical exhaustion, among other factors [17].

The two main hypotheses proposed for the physiological origin of acute attacks are the potential neurotoxicity of ALA/PBG accumulation, or heme deficiency leading to decreased hemoprotein function [18] and energy production in the mitochondria. The first is the most widely accepted hypothesis, as the onset of acute attacks has always been associated with the accumulation of porphyrin precursors. Specifically, symptomatology has been attributed to ALA because (i) patients with other diseases, such as hereditary type I (OMIM 276700), lead poisoning, or the ultrarare hepatic ALAD deficiency porphyria (Table 1), are associated with neurological diseases similar to acute attacks but only accumulated ALA; (ii) in vitro assays confirm the association of ALA with oxidative stress; (iii) ALA selectively competes for the binding of γ-aminobutyric acid (GABA) to synaptic GABA receptors in the postsynaptic membrane of neurons [19]. Furthermore, some authors suggest that polymorphisms in peptide transporter 2 (PEPT2), particularly PEPT2 * 1 * 1, greatly increase serum ALA affinity, which could be related to the passage of toxic ALA to the brain through the choroid plexus, and an increased susceptibility to developing neuropsychiatric symptoms [20].

Despite all of this, the relationship between porphyrin precursor levels and prodrome symptoms is still unclear. Different authors have also pointed out that in porphyria, elevation of ALA may be necessary but not sufficient for the development of an acute attack [21], since ALA administration in a male volunteer [22] and in mice [23] did not produce acute symptoms. In fact, a large urinary loss of liver succinyl-CoA and glycine (used for the production of ALA and PBG) during the acute porphyria attack supports the hypothesis of a profound, although reversible, impact of acute attack on mitochondrial energy metabolism [24]. More recently, preventive treatment with experimental liver-targeted insulin (the fusion protein of insulin and apolipoprotein A-I, Ins-ApoAI) in AIP mice improved pain and motor coordination although excretion of ALA and PBG remained high [25]. This insulin-ApoAI showed an increased serum half-life and high hepatic tropism compared to unconjugated insulins, which improved the mobilization of adipose tissue energy stores and increased hepatocyte glucose uptake [26]. In addition to increasing the energy supply to the liver of porphyria mice, the ApoAI component induced mitochondrial biogenesis [27], which secondarily protected against the porphyrinogenic effects of phenobarbital administration [25]. These data support that low energy production, caused by cataplerosis of the TCA cycle and the reduced availability of energy metabolites during acute attacks, could play a role in modulating the severity of porphyria attacks.

## 3. Current Treatments

Current treatments are based on down-regulation of hepatic *ALAS1* expression using carbohydrate loading, intravenous (iv) hemin therapy, or the subcutaneous (sc) administration of a small interfering RNA (siRNA) targeting *ALAS1* mRNA. The frequency and severity of acute attacks determine the classification of patients into different groups, which can condition their treatment.

Latent porphyria

After a first attack, the precipitating agent is identified and, if possible, removed. Although most patients may not experience an acute attack again, they may maintain high urinary excretion of ALA and PBG for years and are called Asymptomatic High Excretors (ASHE). These patients do not receive treatment, although a recent study shows that 46.4% report chronic symptoms associated with porphyria, such as abdominal pain, fatigue, muscle pain, and insomnia [28].

2.Patients suffering sporadic acute attacks (1 to 3 per year)

Symptoms are very heterogeneous and include autonomous (intense pain typically in the abdomen that can also affect the back, legs, arms, or chest; nausea; vomiting; diarrhea/constipation; hypertension; and/or tachycardia), central (seizures, anxiety, depression, reduced consciousness, psychosis, insomnia, hallucinations or posterior reversible encephalopathy syndrome (PRES) on MRI scan, among others) and peripheral (muscle weakness, paralysis, reduced tendon reflexes) nervous system involvement. Severe neurological complications may cause death due to respiratory and bulbar paralysis [29,30,31]. Acute attacks can be classified according to their clinical severity:

2.1.Mild pain and no paresis

Carbohydrate overload (300 to 500 g/day, based on oral or iv glucose infusions) is recommended for the treatment of these patients. Hemin therapy (3–4 mg/kg/day, iv hemin arginate, Normosang^®^ in Europe and lyophilized hematin, Panhematin^®^ in the US, both from Recordati, Milan, Italy) is more effective than glucose in reducing the formation of porphyrin precursors but is more expensive and is not available in all countries. Hemin therapy acts through retroinhibition of the ALAS1, which reduces production and accumulation of PBG and ALA (Figure 2). Hemin treatment lasted from one to four days, and biochemical remission of ALA and PBG is typically not produced until two or three days after the beginning of treatment. The reduction in abdominal pain is typically observed on the third day of treatment [32].

2.2.Severe attacks

A mild attack can quickly become a severe attack, characterized by severe neuropathic abdominal and muscle pain, significant hyponatremia, urinary retention or incontinence, peripheral neuropathy (85% of sporadic AIP), or central nervous system (CNS) involvement. The treatment recommended for severe porphyria attacks consists of daily administration of hemin for a period of 4 days (3–4 mg/kg of hemin/day). The efficacy of the treatment is very difficult to assess due to significant variability among patients and the low number of patients included in each treatment group. In a review conducted by The American Porphyria Foundation in which all porphyria cases published between 1976 and 2004 were included (71 publications involving approximately 1000 patients), the main conclusion was that the efficacy of hemin therapy depends on the early initiation of iv administration of this compound [33]. In a double-blind, placebo-controlled trial conducted with 12 patients in which hemin treatment was delayed for two days, no statistical benefit was associated with hemin treatment [34].

2.3.Patients suffering frequent acute attacks (≥3 attacks per year)

This group of AIP patients represents approximately 5% of symptomatic patients, mainly women (80%). This group of patients usually requires hospitalization and experiences chronic symptoms that adversely influence daily functioning and undermine their quality of life [35,36]. Chronic opiate therapy is often needed to control pain [37]. Furthermore, these patients are chronically exposed to the potential toxicity of heme precursors when passing through the kidney, especially ALA, which has been identified as being responsible for progressive renal failure [38,39], or to an increased risk of developing hepatocellular carcinoma [40,41,42,43], the long-term complications associated with acute hepatic porphyrias.

Of the registered patients who experienced frequent acute attacks (>2 attacks/year) in the USA and the EU, 75% had 3.5 attacks per year and were treated with four doses of hemin per attack, while the other 25% received one or two doses of preventive hemin per month [36]. Although there are no reports confirming the efficacy of hemin administration in preventing acute attacks, off-label administration of prophylactic iv heme infusions is commonly used [44]. An audit report by the National Acute Porphyria Service in England concluded that prophylactic hemin arginate appears to be beneficial in patients with recurrent acute porphyria symptoms, but more studies are required to support its use [44]. However, repeated administration of hemin therapy can cause unwanted effects and complications, such as thrombophlebitis at the peripheral vein infusion site (requiring administration through a central vein), iron overload (each 250 mg dose of hemo contains 22.7 mg of iron), or induction of the HO-1 enzyme, causing the reduction of the regulatory free heme level in hepatocytes [45]. This situation re-induces the regulatory feedback mechanism of heme in cells through the activation of *ALAS1* expression, reducing the therapeutic efficacy of hemin administration over time [45]. Finally, hemin therapy is not recommended in patients since administration of large amounts of hemin has been associated with transitory renal failure [46].

Recently, sc administration of an ALAS1 siRNA (givosiran, givlaari^®^, Alnylam Therapeutics, Cambridge, MA, USA) has been approved for the treatment of severely affected patients who experience recurrent porphyria acute attacks (Figure 2). Givosiran therapy is based on the fact that the accumulation of ALA and PBG precursors is the sole cause of the pathophysiology of the disease. The Phase III clinical trial (NCT03338816) and a 24 month interim analysis of efficacy and safety have shown good results in preventing ALA/PBG accumulation and reduced the frequency of acute attacks by 87%. Although givosiran had an acceptable safety profile and was generally well tolerated in patients with acute hepatic porphyrias in clinical studies [47], adverse events (AE) (90% vs. 80%), severe AEs (17% vs. 11%), and serious AEs (21% vs. 9%) were more common with givosiran than with placebo in the ENVISION trial [47]. Among AEs, injection site reactions (25% vs. 0%), nausea (27% vs. 11%), chronic kidney disease (10% vs. 0%), decreased estimated Glomerular Filtration Rate (eGFR) (6% vs. 0%), rash (6% vs. 0%), increased levels of alanine transaminase (8% vs. 2%), and fatigue (10% vs. 4%) were more frequent among the patients receiving givosiran than in the placebo group. Interpreting the safety data is complicated by the fact that chronic kidney disease and liver damage are common coexisting illnesses and long-term complications of acute hepatic porphyria [38,39,41,48].

Hepatic targeting of givosiran is mediated by interaction of *N-acetylgalactosamine* linked to siRNA with the asialoglycoprotein receptor (ASGPR). However, ASGPR is also expressed in renal tubular cells, and its ligation could be related to changes in serum creatinine and kidney function complications, assessed by decreased eGFR [47]. A recent report of patients followed for two years concluded that givosiran is associated with a moderate transient increase in serum creatinine without signs of kidney injury. However, the long-term deleterious impact of *ALAS1* inhibition on renal function cannot be ruled out [49].

Interestingly, abdominal pain was more common as an adverse effect in patients receiving givosiran than in those given placebo. Since these patients exhibited a substantial reduction in ALA levels, this observation could indicate that, in addition to excess ALA, other factors may also play a role in this clinical characteristic. Given that givosiran introduces a second block in the pathway, it cannot be excluded that a further reduction in heme availability could hamper the activation of multiple biological processes that take place in the liver in situations of stress (antioxidant response, inflammation, hypoxia), important detoxification processes, or adequate energy supply for natural hepatocyte proliferation or liver regeneration. Therefore, reduced availability of heme in the liver could be associated with the reported AEs associated with ALAS-1 iRNA, such as bile acid disorders [50], reduced drug metabolization rates [51,52], or dysregulation of one-carbon metabolism [53,54]. Concomitant hypermethioninemia and hyperhomocysteinemia resembling classic homocystinuria have been associated with givosiran treatment. These are likely to be attributable to an impairment in the trans-sulfuration pathway catalyzed by cystathionine β-synthase. This enzyme uses vitamin B6 as a cofactor [55] and S-adenosylmethionine as an allosteric activator of enzyme activity [54].

Heme is not only required for the synthesis of hemoproteins (such as the cytochromes of the mitochondrial respiratory chain and those involved in drug metabolism); it is also essential as a substrate for inducible HO-1. This enzyme converts heme to biliverdin and carbon monoxide. The latter is a potent vasodilatory, anti-inflammatory, and immunomodulatory agent. Thus, low HO activity resulting from decreased substrate availability would increase vulnerability to pro-inflammatory insults in the liver [56]. Given that ASGPR is also expressed in peripheral monocytes, peritoneal macrophages, endometrium, placenta, and renal tubular cells, givosiran could affect the anti-inflammatory activity of these cells. Thus, biodistribution studies of givosiran in humans would be helpful in evaluating its presence in extrahepatic tissues.

In summary, available evidence indicates that givosiran is an efficient therapeutic option to prevent acute attacks with recurrent porphyria in severely affected patients. However, there is still room for improvement in AIP therapy to cover the full spectrum of the disease, from sporadic to recurrent attacks, regardless of their severity; and to prevent the appearance of the AEs and severe AEs associated with recurrent administrations of givosiran.

## 4. Innovative Therapies

Current approved treatments for acute porphyria do not provide an etiological solution to the disease. The current prevalent R&D trends related to AIP therapy focus on increasing hepatic PBGD activity (Figure 2 and Figure 3) and restoring the physiological regulation of the heme synthesis pathway.

1. Oral administration of pharmacological chaperones with the aim of prolonging the half-life of a mutated protein [57]. However, this approach is in the preclinical stage and may only be applied in those patients in whom the PBGD mutation produces a partially active protein with a short half-life.

2. Subcutaneous administration of enzyme replacement treatment could be an option to reduce porphyrin precursors during an acute attack, as demonstrated in a murine model (Figure 2). However, a clinical trial with an rh-PBGD administered twice a day was discontinued due to the instability of the enzyme in the circulation and a lack of efficacy to protect against acute attacks [58].

Recently, we developed a recombinant PBGD protein conjugated to Apolipoprotein AI (rApoAI-PBGD) that targets the liver [59] (Figure 3). The administration of this fusion protein via iv and sc injection efficiently prevented and abrogated acute attacks in AIP mice [59]. The conjugated PBGD protein had an increased serum half-life (from 45 min to almost 10 h) and persisted in serum for up to 6 days incorporated into high-density lipoprotein (HDL) particles. Furthermore, the administration of a high dose of fusion protein (300 nmol/kg) increased liver PBGD activity, providing rapid protection against the accumulation of porphyrin precursors during an ongoing acute attack [59].

The main advantage of this enzyme replacement therapy (ERT) approach is that it acts in the three compartments involved in the acute attack, liver, serum, and brain. Furthermore, it can be administered by sc injection, thereby being less constraining than iv administration. As tested, rhApoAI-PBGD administration did not lead to any immunological anaphylactic reactions or antibody formation in preclinical studies [59], which are typical limitations of protein-based ERT. However, recombinant proteins have a high production cost and the design of large-scale production processes is required.

A prokaryotic cell factory was used for proof of concept (PoC) in AIP mice due to its fast growth, ease of handling, and cost-effectiveness [60]. However, antibiotic resistance to clone selection and the use of the polyhistidine tag at the N-terminal region to facilitate the purification process are not recommendable for obtaining Good Manufacturing Practice (GMP) requirements. The large-scale production of rApoAI-PBGD was also assessed in the Chinese Hamster Ovary (CHO)-K1SP cell line as eukaryotic host. The histidine tag was replaced by a signal peptide sequence that allows the protein to be secreted into the supernatant. Although this cell line increased the production yield by a factor of 4 × 10^6^ (50–60 mg of rhApoAI-PBGD per L of supernatant), 99% of the production was obtained in the form of aggregates. In fact, protein aggregates were ineffective in protecting against acute attack in AIP mice. Further studies are required to optimize the rhApoAI-PBGD sequence to avoid aggregate formation.

3. Gene therapy (GT) aiming at a long-term supply of PBGD protein to hepatocytes represents a promising therapeutic option for those patients with AIP who experience frequent and severe acute attacks, which require repeated administration of hemin or liver transplantation (Figure 3). Liver-directed GT can prevent the occurrence of acute crises and neurologic complications and therefore may improve quality of life and reduce hospitalization and health care costs.

Liver-directed GT using a vector rAAV2/5 encoding human *PBGD* cDNA under the control of a liver-specific promoter (rAAV2/5-PBGD) was able to restore liver PBGD activity to average values and protected AIP mice against the accumulation of porphyrin precursors after phenobarbital challenge [61]. PoC in AIP mice confirmed that overexposure to PBGD protein was safe, and partial recovery of liver PBGD levels was sufficient to prevent acute crises. However, in a clinical trial (NTC020082860), this strategy failed to reduce the levels of porphyrin precursors due to insufficient liver transduction at the doses tested (up to 1.8 × 10^13^ genome copies/kg) [62]. Ongoing efforts are focusing on improving the efficacy of AAV-GT vectors [63,64].

4. mRNA-based therapies formulated in lipid nanoparticles (LNPs) are starting to be used in clinical trials of inborn metabolism errors [65] and, also preclinically, in acute porphyrias [66,67]. LNPs are multicomponent spherical vesicles (~100 nm) consisting of phospholipids, cholesterol, polyethylene glycol (PEG) conjugated lipids, and mainly amino lipids [68,69,70]. Intracellular delivery and endosomal escape are the main challenges for the success of therapeutic mRNA and, together with tolerability, are addressed by the amino lipid components [71]. Phospholipids also aid in endosomal escape and mRNA release into the hepatocyte cytosol by providing fusogenicity with the target cell membranes [71]. Cholesterol improves particle stability by modulating membrane integrity and stiffness, and PEG inhibits interactions with plasma proteins allowing greater circulation time and escape from phagocytic cells [72]. The natural tropism of these LNPs for the LDL receptor has facilitated applications focusing on the liver [73].

The iv administration of hPBGD mRNA formulated in LNPs (Figure 3) showed a rapid increase in PBGD activity in hepatocytes and a fast normalization of urine porphyrin precursors in ongoing attacks in AIP mice [66]. In fact, hPBGD mRNA maintained the same degree of protection after repeated administrations. Rapid effect is a very important aspect in acute presentations of the disease, while sustained efficacy after repeated dosing is necessary for chronic presentations. Proven safety and translatability after multiple administrations would allow mRNA formulated in LNPs and rhApoAI-PBGD ERT to treat chronic presentations, where re-administration and individual dosage decisions could be taken according to the clinical and biochemical status of the patient. These features are crucial for a new era of personalized and precision medicine.

## 5. Effect of Increased Hepatic PBGD on Poorly Described Disease Parameters Associated with Acute Hepatic Porphyrias

Ventricle enlargement and reduced perfusion of the brain in the central nervous system. Among CNS-related abnormalities, structural changes and brain perfusion have been poorly characterized in patients with acute liver porphyria. Recently, neuroradiological studies have described an enlargement in the brain ventricles in eight patients with severe AIP and recurrent acute attacks without posterior reversible encephalopathy syndrome [74]. In addition, a decrease in brain perfusion during acute attacks was also reported in two of the patients in whom perfusion imaging data were acquired [74].

The effect of therapy to increase liver PBGD levels in response to these brain changes was studied in the AIP mouse model [74]. AIP mice developed chronic brain ventricle dilatation even in the presence of slightly increased porphyrin precursors. In fact, enlargement was exacerbated after repeated phenobarbital challenges that caused increased ALA and PBG accumulation. However, a direct neurotoxic effect of ALA and PBG is an unlikely factor because porphyrin precursors accumulate markedly in their liver and plasma, but less in the CNS as previously reported in AIP mice by Yasuda et al. [75], and in a variegate porphyria (VP) model in rabbits described by Jericó et al. [67]. In this study, a rAAV2/5-*PBGD* vector was used to obtain sustained hepatic PBGD over-expression in the liver of AIP mice for several months after a single injection [61]. It is noteworthy that vectors with hepatic tropism, such as PBGD mRNA formulated in LNPs, could obtain the same effect as long as an adequate dose is found.

AIP mice not exposed to phenobarbital challenge also exhibited reduced cerebral blood flow with normal systolic blood pressure [74]. Of interest, correction of liver PBGD deficiency reversed brain perfusion reduction and prevented progressive brain ventricle enlargement in AIP mice subjected to repeated acute porphyric attacks (Table 2) [74]. The authors suggested that constitutive small vessel dysfunction associated with the systemic arterial hypertension that occurs during acute porphyria attacks could cause an imbalance in cerebrospinal fluid resulting in brain ventricle enlargement. Similarly, a previous report showed increased local vasoconstrictor responses in the mesenteric arteries in this mouse model with significant vasodilation after hemin administration [76].

An increase in systolic blood pressure associated with acute attacks was described in AIP mice [66] and in a VP model in rabbits [67]. The latter is a pharmacological model developed in a large animal by administering two porphyrinogenic drugs (2-allyl-2-isopropylacetamide (AIA) and Rifampicin) [67], which reproduces the characteristics of acute attacks in the context of VP (the second most frequent acute hepatic porphyria after AIP) [77]. Notably, hemin administration in both AIP mice and VP rabbits protects against increased systolic blood pressure, although recurrent administration did not effectively protect against the accumulation of porphyrin precursors. Therefore, these data suggest that hypertension might be related to heme availability during porphyria attacks rather than the accumulation of porphyrins and porphyrin precursors that persist after hemin therapy in both of these animal models.

Regarding liver status, VP rabbits also reproduced some of the signs and symptoms associated with acute attacks of porphyria, such as inflammation, oxidative stress, and altered energy supply caused by disturbed OXPHOS activities [67] (Table 2). VP rabbits showed overexpression of the *Ho-1*, *Hsp70,* and *Hepcidin* genes, biomarkers of inflammation and oxidative stress. VP rabbit urine also showed increased excretion of urinary thiobarbituric acid reactive substances (TBARS), which is a biomarker of lipid peroxidation, and is highly correlated with urinary ALA accumulation. mRNA-based therapy showed more efficient protection against inflammation and oxidative stress than hemin [67].

Finally, porphyria models also showed disturbed energy supply, as AIP mice exhibit altered glucose metabolism and glucose tolerance tests (GTT) during fasting [25,67,78,79] and showed a reduced oxygen consumption rate (OCR) in the liver [66]. In VP rabbits, the liver activity of the mitochondrial respiratory chain complex was also reduced [67]. Administration of hPBGD mRNA was able to counteract all these biochemical disturbances in both models, while treatment with hemin only partially protected against OCR in AIP mice (Table 2).

**Table 2 life-12-01858-t002:** Summary of symptoms and characteristics of human patients with severe AIP and experimental models of acute liver porphyria. The color code represents the degree of protection against a particular symptom of hepatic augmentation PBGD therapy in experimental models. Green, orange, and red indicate effective, partial, or no protection offered by the therapy, respectively.

Human Patients with Severe AIP		Experimental Model		Therapy	
				Augmenting Hepatic PBGD (rAAV-Mediated GT)	
Central NervousSystem	Ventricle enlargement [74]	AIP mouse	Present; Exacerbated after recurrent attacks	Effective protection but not reverse alterations previous to therapy	
	Reduced brain perfusion during the acute attack [74]		Present	Effective protection	
				Augmenting hepatic PBGD (mRNA therapy)	Hemin
Peripheral Nervous System	Motor impairment [2,3]	VP rabbit	Present	Effective protection	Partial protection
Autonomic Nervous System	Chronic hypertension [2,3]		Present	Effective protection	Effective protection
Liver metabolism	Altered glucose homeostasis [25]		Present	Effective protection	No protection
Liver function	Lipid peroxidation [2,3]		Present	Effective protection	Partial protection
	Inflammation [15]		Present	Effective protection	Partial protection
	Cytoplasmic Stress [80]		Present	Effective protection	Exacerbated
	Altered hemoprotein function [66]		Reduced mitochondrial respiratory chain	Effective protection	No protection

## 6. Conclusions

Based on the PoC obtained in animal models, increasing PBGD levels appears to be a promising strategy for the etiological treatment of AIP, regardless of whether it is achieved by the administration of a recombinant rApoAI-PBGD protein, an rAAV-mediated GT, or mRNA encapsulated in LNPs. Indeed, a single administration of the rApoAI-PBGD protein or PBGD-mRNA induced a rapid and efficient overexpression of a functional PBGD protein in the liver of mice and large animals. Therefore, these approaches can be used to treat ongoing acute attacks. However, further studies are needed to optimize the AAV vector and large-scale production of the rApoAI-PBGD protein.

The messenger RNA technology is being successfully tested in several clinical trials for five metabolic diseases (NCT 05095727: Glycogen storage disease Type 1A; NTC 04574830: Glycogen storage disease Type 3; NCT05130437: Propionic Acidemia; NCT 04899310: Methylmalonic Acidemia; NTC 04442347: Ornithine transcarbamilase deficiency) so that experience could quickly be transferred to patients with porphyria. Furthermore, this product could be applied to all patients regardless of clinical course, both chronic and sporadic presentations, as well as ASHE until normalization of the urinary excretion of porphyrin precursors.

## Figures and Tables

**Figure 1 life-12-01858-f001:**
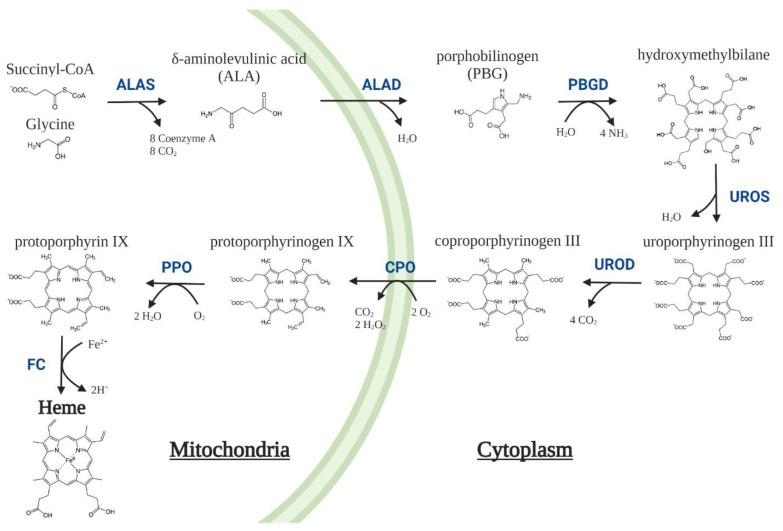
Heme biosynthesis pathway in mammalian cells. The pathway involves eight enzymes, four located in the cytoplasm and the other four in the mitochondria. The first and limiting step is the decarboxylative condensation of the non-essential amino acid glycine with succinyl-CoA coming from the tricarboxylic acid (TCA) cycle, which is catalysed to form δ-aminolevulinic acid (ALA) by ALA-synthase (ALAS). Eight ALA molecules are required to synthesize four porphobilinogen (PBG) molecules that, subsequently, are used for the synthesis of a single hydroxymethylbilane molecule that is promptly converted to cyclic tetrapyrroles known as porphyrinogens, and finally to a heme prosthetic group [5]. ALAD = δ-aminolevulinic acid dehydratase; PBGD = porphobilinogen deaminase; UROS = Uroporphyrinogen III synthase; UROD = Uroporphyrinogen III decarboxylase; CPO = coproporphyrinogen oxidase; PPO = protoporphyrinogen oxidase; FC = ferrochelatase.

**Figure 2 life-12-01858-f002:**
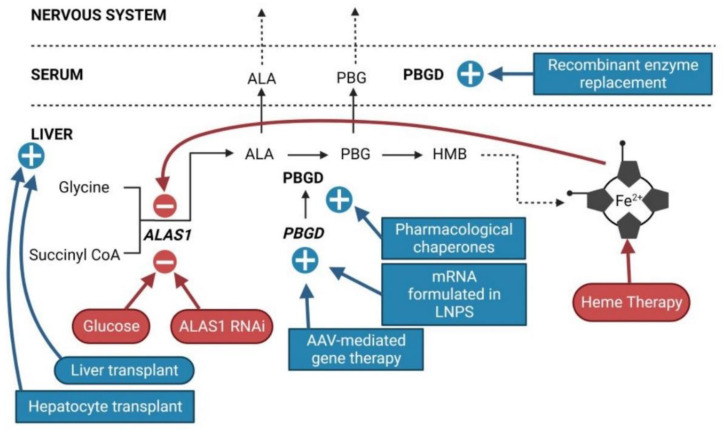
Sites of action of current and innovative therapeutic options for AIP. Current approved therapies are represented in round outlines. Emerging innovative therapies are represented in square outlines. Therapies that inhibit *ALAS1* are in red. Therapies that increase PBGD activity are in blue.

**Figure 3 life-12-01858-f003:**
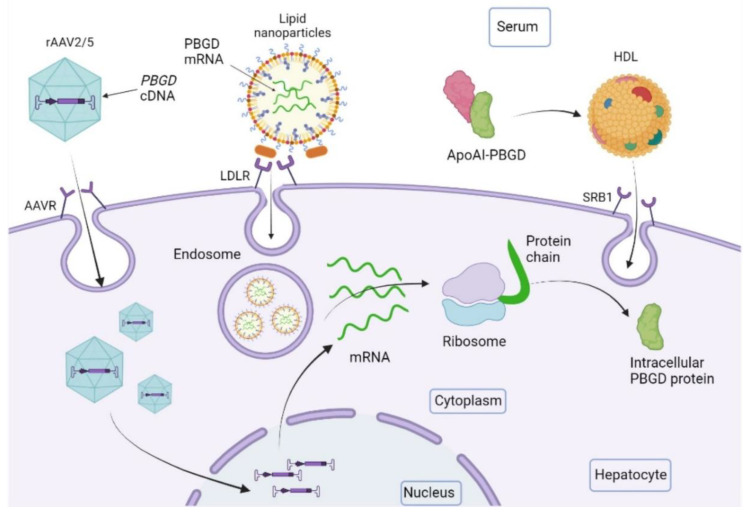
Outline of the performance of new emerging therapies for increasing liver PBGD levels. The recombinant adeno-associated virus 2/5 (rAAV2/5) is the vector used for gene therapy studies. It contains the PBGD complementary DNA sequence (*PBGD* cDNA) and targets the liver through specific receptors (AAVR). PBGD messenger RNA (PBGD mRNA) formulated in lipid nanoparticles is incorporated into hepatocytes through LDL receptors. The recombinant protein PBGD linked to apolipoprotein AI (ApoAI-PBGD) is transported by HDL particles and internalized in hepatocytes by the centripetal transport of cholesterol through the class B type 1 (SRB1).

**Table 1 life-12-01858-t001:** Type of porphyria associated with the abnormalities in each specific enzyme in the heme synthesis pathway. Porphyrias are characterized by loss-of-function mutations in any of the last seven enzymes, except X-linked Protoporphyria, which is associated with gain-of-function mutations in erythroid ALAS2. In contrast, ALAS2 deficient activity is associated with X-linked sideroblastic anemia (OMIM 300751) [6,7]. Pathogenic mutations in the housekeeping ALAS1 gene have not been reported. Hepatoerythropoietic porphyria refers to a homozygous form of PCT, which has a childhood onset. GoF = gain-of-function mutation. LoF = loss-of-function mutation.

Enzyme	Mutation	Disease	OMIM
δ-Aminolevulinic acid synthase 2 (ALAS2, EC 2.3.1.37)	GoF	X-linked Protoporphyria (XLP)	300752
δ-Aminolevulinic acid dehydratase (ALAD, EC 4.2.1.24)	LoF	ALAD Deficiency Porphyria (ADP)	612740
Porphobilinogen deaminase (PBGD, EC 2.5.1.61)	LoF	Acute Intermittent Porphyria (AIP)	176000
Uroporphyrinogen III synthase (UROS, EC 4.2.1.75)	LoF	Congenital Erythropoietic Porphyria (CEP)	263700
Uroporphyrinogen III decarboxylase (UROD, EC 4.1.1.37)	LoF	Porphyria Cutanea Tarda (PCT) Hepatoerythropoietic porphyria (HEP)	176100
Coproporphyrinogen oxidase(CPO, EC 1.3.3.3)	LoF	Hereditary Coproporphyria (HCP)	121300
Protoporphyrinogen oxidase(PPO, EC 1.3.3.4)	LoF	Variegate Porphyria (VP)	176200
Ferrochelatase(FC, EC 4.99.1.1)	LoF	Erythropoietic Protoporphyria (EPP)	177000

## Data Availability

Not applicable.

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
