# Peer review of "Recent Insights into the Pathogenesis of Acute Porphyria Attacks and Increasing Hepatic PBGD as an Etiological Treatment"

_life, 2022, doi:10.3390/life12111858_

Round 1
Reviewer 1 Report
The Authors present a very comprehensive overview of the pathogenesis of acute porphyria attacks and update on emerging treatments.
This review was definitely a major effort and has great educational relevance. Some minor comments:
- some English polishing is necessary (e.g., 'evidence' and not 'evidences')
- I understand this is a narrative review; however, can the Authors briefly comment on how considered papers were screened and selected?
- can the Authors expand on the emerging treatments, and provide their views on the future progresses in the field?
Author Response
REVIEWER 1
The Authors present a very comprehensive overview of the pathogenesis of acute porphyria attacks and update on emerging treatments. This review was definitely a major effort and has great educational relevance.
Some minor comments:
- some English polishing is necessary (e.g., 'evidence' and not 'evidences')
AUTHORS: As suggested, the revised manuscript has been carefully edited by a native English speaker.
- I understand this is a narrative review; however, can the Authors briefly comment on how considered papers were screened and selected?
AUTHORS: We searched PubMed from inception to September 2022 for papers dedicated to current and innovative emerging therapies for acute hepatic porphyrias. The search strategy included the keywords: “hepatic porphyrias”, “heme synthesis”, “interference RNA technology”, “liver gene therapy”, “enzyme replacement therapy targeting the liver”, “pharmacological chaperones”, “heme replacement therapies”. Three authors reviewed and extracted information from each study. We also recovered information from oral and poster communications at the specialized porphyria congress held in September 2022 (ICPP2022 Sofia, Bulgaria, https://icpp2022.com/). This point has been included in the Author Contributions note.
- can the Authors expand on the emerging treatments, and provide their views on the future progresses in the field?
AUTHORS: This point has been added to the abstract of the new version of the manuscript. We acknowledge the reviewer for the comments which allowed us to improve our manuscript.
Reviewer 2 Report
The review “Recent insights into the pathogenesis of acute porphyria attacks and increasing hepatic PBGD as an etiological treatment” by Daniel Jerico et al., is a very interesting one. The authors have provided a timely and detailed updated account of various aspects of the clinical problems associated with the orphan disease acute intermittent porphyria (AIP). Their account also provides an update about the treatment options that are in use currently, and also the ones that have shown promise in experimental animal models and in preliminary human studies to reduce the severity of AIP. It will be very useful for non-specialist readers to get a clear idea about what is AIP, about its pathogenesis, current treatment options, and future possibilities to get better and effective treatment methods. The authors have discussed with evidence from multiple ongoing studies the potential and challenges of future treatment strategies such as, gene therapy, pharmacological chaperones, enzyme replacement therapy, mRNA-based therapies with lipid nanoparticles etc., to cure AIP. These have huge potential if proved successful in larger clinical trials.
Ideally, all therapies against AIP aim to enhance the hepatic activity of PBGD, as this has been more effective in giving protection than administering Hemin. This has been highlighted by the analysis of the recent progress with studies involving both humans and animals. Liver tropism and appropriate functioning of the PBGD enzyme delivered by any mean described in the review will potentially be very helpful. The review has been structured well and the resources cited are updated. This helps the readers to get further information on the recent progress in the field.
Minor modification is needed to the text where abbreviations are not expanded in full at the point of first use. PGC1a in line 123, ApoAI in line 152, AHP in line 239, ALT in line 243, HDL in line 311, PoC in line 322, PRES in line 384 and GTT in line 428 etc., need to be expanded in full for the ease of readers.
The review is well written and well analysed.
Author Response
REVIEWER 2
Comments and Suggestions for Authors
The review “Recent insights into the pathogenesis of acute porphyria attacks and increasing hepatic PBGD as an etiological treatment” by Daniel Jerico et al., is a very interesting one. The authors have provided a timely and detailed updated account of various aspects of the clinical problems associated with the orphan disease acute intermittent porphyria (AIP). Their account also provides an update about the treatment options that are in use currently, and also the ones that have shown promise in experimental animal models and in preliminary human studies to reduce the severity of AIP. It will be very useful for non-specialist readers to get a clear idea about what is AIP, about its pathogenesis, current treatment options, and future possibilities to get better and effective treatment methods. The authors have discussed with evidence from multiple ongoing studies the potential and challenges of future treatment strategies such as, gene therapy, pharmacological chaperones, enzyme replacement therapy, mRNA-based therapies with lipid nanoparticles etc., to cure AIP. These have huge potential if proved successful in larger clinical trials.
Ideally, all therapies against AIP aim to enhance the hepatic activity of PBGD, as this has been more effective in giving protection than administering Hemin. This has been highlighted by the analysis of the recent progress with studies involving both humans and animals. Liver tropism and appropriate functioning of the PBGD enzyme delivered by any mean described in the review will potentially be very helpful. The review has been structured well and the resources cited are updated. This helps the readers to get further information on the recent progress in the field.
Minor modification is needed to the text where abbreviations are not expanded in full at the point of first use. PGC1a in line 123, ApoAI in line 152, AHP in line 239, ALT in line 243, HDL in line 311, PoC in line 322, PRES in line 384 and GTT in line 428 etc., need to be expanded in full for the ease of readers.
The review is well written and well analysed.
AUTHORS: We thank the reviewer for these comments. Minor modifications have been made as suggested. “AHP”, “ALT”, and “PRES” have been expanded, but the other abbreviations have been removed because the terms were only used once in the manuscript.
